# Evaluation of the Probiotic In Vitro Potential of Lactic Acid-Producing Bacteria from Canine Vagina: Possible Role in Vaginal Health

**DOI:** 10.3390/ani12060796

**Published:** 2022-03-21

**Authors:** Brian Morales, Livia Spadetto, Maria Àngels Calvo, Marc Yeste, Leonardo Arosemena, Teresa Rigau, Maria Montserrat Rivera del Alamo

**Affiliations:** 1Department of Animal Health and Anatomy, Faculty of Veterinary Medicine, Universitat Autònoma de Barcelona, 08193 Bellaterra, Spain; brianmorales2@hotmail.com (B.M.); mariangels.calvo@uab.cat (M.À.C.); estebanleonardo.arosemena@uab.cat (L.A.); 2Toxicology Research Group, Faculty of Veterinary Medicine, University of Murcia, Campus Espinardo, 30100 Murcia, Spain; livia.spadetto@um.es; 3Department of Biology, Faculty of Sciences, University of Griona, 17071 Girona, Spain; marc.yeste@udg.edu; 4Department of Animal Medicine and Surgery, Faculty of Veterinary Medicine, Universitat Autònoma de Barcelona, 08193 Bellaterra, Spain; teresa.rigau@uab.cat; 5Fundació Hospital Clínic Veterinari, Faculty of Veterinary Medicine, Universitat Autònoma de Barcelona, 08193 Bellaterra, Spain

**Keywords:** canine vagina, probiotic, *Lactobacillus* spp., gastrointestinal conditions, antimicrobial activity

## Abstract

**Simple Summary:**

The use of microorganisms with probiotic capacity, obtained from the natural microbiota itself, is an alternative tool for the treatment and prevention of some animal pathologies. In this study, a total of 94 bitches were sampled in order to identify the lactic acid-producing microbiota of the vagina. Following the general isolation of the vaginal microbiota, 100 strains of lactic acid-producing bacteria (LAB) were identified. Through Gram staining and basic biochemical tests (catalase, oxidase and haemolysis), 13 LAB strains with possible probiotic capacity were selected to undergo an evaluation of resistance to gastrointestinal conditions (pH, lysozyme, bile salts and hydrogen peroxide) and safety and efficacy in vitro (resistance to antibiotics and antimicrobial capacity by disk diffusion). After the selection, only 3 of 100 strains showed in vitro probiotic potential.

**Abstract:**

Lactic acid-producing bacteria (LAB) are being widely studied due to their probiotic potential. The aim of the present study was to determine and identify the presence of LAB from canine vaginal samples, as well as to evaluate their probiotic in vitro potential. Ninety-four bitches were included in the study. Vaginal samples were obtained by means of a sterile swab and streaked on Man Rogosa Sharpe agar plates. A total of 100 LAB strains were obtained and submitted to Gram stains and basic biochemical tests, which included catalase, oxidase and haemolysis tests. Thirteen strains belonging to the genera *Lactobacillus* (n = 10), *Lactococcus* (n = 2) and *Pediococcus* (n = 1) were selected as potential probiotics and further subjected to evaluation of resistance to gastrointestinal conditions (pH, lysozyme, bile salts and hydrogen peroxide) and safety and efficacy in vitro (resistance to antibiotics and antimicrobial capacity). Only three strains, one *Lactobacillus lactis* and two *Lactobacillus plantarum,* accomplished the requirements for being considered as potential in vitro probiotics.

## 1. Introduction

Lactic acid-producing bacteria (LAB) are a group of Gram-positive bacteria, strictly anaerobic or aerotolerant, usually immotile, nonsporulating and not pigmented, catalase- and oxidase-negative, that have the form of bacilli or cocci [1,2]. LAB include the genera *Aerococcus*, *Carnobacterium*, *Enterococcus*, *Tetrageonococcus*, *Vagococcus*, *Lactobacillus*, *Pediococcus*, *Leuconostoc*, *Oenococcus*, *Weisella*, *Lactococcus* and *Streptococcus,* which belong to the order *Lactobacillales* of the phylum Firmicutes, and the genus *Bifidobacterium,* that, in turn, pertains to the order *Bifidobacteriales* of the phylum Actinobacteria [2,3].

Studies focused on vaginal LAB are scarce in the bitch [4,5], despite their role having been thoroughly described in women (see [6] for a review). LAB, specifically *Lactobacillus* spp., are predominant in the vaginal microbiota of fertile women and play a fundamental role in the homeostasis of the reproductive tract, thus contributing to the prevention of urogenital diseases [7,8,9,10]. The main mechanisms involved in this protective role are biofilm formation, which would promote the adhesion and permanence of the bacteria on the epithelium, competitive exclusion of pathogenic microorganisms, production of antimicrobial substances and modulation of the immune response in the host [7,11,12,13,14].

On the other hand, as it has been already stated, Lactobacilli are the most prevalent microorganisms in the women’s vaginal environment from puberty until menopause, when they are depleted [15]. This depletion during menopause has been related to the reduction in oestrogen production, which suggests a hormonal influence on the prevalence of vaginal LAB [16]. Likewise, with the onset of menopause, the vaginal pH increases and returns to neutrality, thus allowing Staphylococci and other skin microorganisms to colonize the vaginal mucosa. Consequently, the composition of the vaginal microbiota gradually resembles that of the cutaneous microbiota [17].

The specific characteristics of LAB have promoted their use as common components of probiotic preparations [18]. The current definition of probiotic organisms proposed by the Food and Agriculture Organization (FAO) and by the World Health Organization (WHO) is “living microorganisms which, when administered in adequate quantities, confer a health benefit to the host” [19].

The potential benefits of probiotic organisms are explained through their mechanisms of action [20], which include (1) the competition with pathogenic microorganisms for nutrients and adhesion areas; (2) the production of antimicrobial substances such as organic acids, hydrogen peroxide and bacteriocins, which inhibit both Gram-positive and Gram-negative bacteria; and (3) the production of fatty acids that reduce pH, which is detrimental for pathogenic bacteria, among others.

For a strain to be considered probiotic, it must undergo a selection process that confirms its safety, that is, specifications of the origin of the strain, absence of pathogenicity and its characteristics of antibiotic resistance, as well as its viability and persistence in the gastrointestinal tract, immunomodulation and its antimutagenic properties [21]. It is important to remark that the individual characterization of a strain as a potential probiotic is mandatory since probiotic properties cannot be extrapolated to a genus or species [19].

Against this background, the aim of the present study was to determine the probiotic in vitro potential of vaginal LAB from healthy bitches by evaluating their microbial safety, microbial resistance to gastrointestinal conditions and antimicrobial capacity.

## 2. Materials and Methods

### 2.1. Sample Collection

In the present study, 94 bitches of different breeds and ages, ranging from seven months to 10 years, were included. All bitches were patients of the Hospital Clínic Veterinari at the Universitat Autònoma de Barcelona (Bellaterra, Spain). Females were distributed in four different groups, namely, neutered (n = 25), heat (n = 48), pregnant (n = 5) and presenting vaginitis (n = 16). All females went through a routine general examination and anamnesis before sampling. As a specific requirement, neutered bitches had to be healthy and were neutered for at least two months before sampling to avoid hormonal effects. In addition, no history of medication during the previous two months was also mandatory.

Vaginal samples were obtained by means of sterile swabs with transport medium (Invasive sterile Eurotubo^®^ Collection swab, Deltalab, Rubí, Spain). Swabs were placed in the transport medium and immediately taken to the laboratory for processing. Microbiological analyses were performed following the protocols established by the Laboratory of Applied and Environmental Microbiology, Department of Animal Health and Anatomy, Universitat Autònoma de Barcelona (Bellaterra, Cerdanyola del Vallès, Spain).

The procedure followed the guidelines of the Ethical Committee Animal Care and Research, Universitat Autònoma de Barcelona (CEEAH 4095). In addition, all manipulations were performed with signed consent from the owners.

### 2.2. Streak, Isolation and Identification

Swabs were streaked directly onto Man Rogosa Sharpe (MRS) (Liofilchem srl, Roseto degli Abruzzi, Italy) agar Petri dishes following a continuous streak method. The plates were then incubated for 24 h at 37 °C ± 2 °C at different respiratory conditions: aerobiosis, anaerobiosis and microaerophilia (5% CO_2_). After that, a first counting of the colonies was performed. The plates in which the growth of colonies had not occurred were incubated for another 24 h.

Plates to be incubated under anaerobic conditions were placed in an anaerobic jar with the Anaerocult^TM^ A system (Merck KGaA, Darmstadt, Germany). The jar was incubated at 37 °C for 48 h. After incubation in MRS media, LAB colonies were counted and isolated using the quadrant streaking technique.

Once colonies were isolated, staining with Gram and Malachite green was performed to select Gram-positive, cocci- or rod-shaped, non-sporulating bacteria. Next, isolated LAB underwent catalase, oxidase and haemolysis tests. Only those bacteria showing negative results for catalase and oxidase were selected. Furthermore, bacteria showing β-haemolysis were also discarded, prioritising those showing γ-haemolysis.

Selected bacteria were then subjected to identification using the API^®^ 50 CHL + 50 CHL Medium^®^ miniaturized systems (Biomérieux S.A., Marcy-l’Étoile, France). API^®^ 50 CHL is used to identify the genus *Lactobacillus* and its nearby microorganisms. By means of an API^®^ 50 CHL Medium ampule, a two-dilution step with McFarland was performed using colonies from a pure culture. Immediately, the suspension of the API^®^ 50 CHL Medium was distributed in the gallery, inoculating only the tubes and coating the tests with paraffin oil. At the end of the inoculation, it was incubated at 36 °C ± 1 °C in aerobiosis for 48 h ± 2 h. Once the incubation time was over, the reading and interpretation of the gallery was performed by means of the Apiweb^TM^ Identification Software (Biomérieux S.A., France).

### 2.3. Evaluation of the Probiotic Potential

To evaluate the in vitro probiotic potential, the selected strains underwent three different steps: evaluation of microbial safety, determination of microbial resistance to gastrointestinal conditions and assessment of antimicrobial capacity.

#### 2.3.1. Evaluation of Microbial Safety

In order to evaluate the in vitro sensitivity of LAB to antibiotics, the selected strains were tested against the nine specific antimicrobials established by the “Guidance on the assessment of bacterial susceptibility to antimicrobials of human and veterinary importance” [22]. According to these criteria, bacteria with probiotic potential have to be sensitive to ampicillin, vancomycin, gentamicin, kanamycin, streptomycin, erythromycin, clindamycin, tetracycline and chloramphenicol.

The assessment of microbial safety was performed by means of an antimicrobial susceptibility disk diffusion test (Oxoid, Basingtoke, UK) on Mueller Hinton agar plates (Biolifes.r.l., Milan, Italy). For that purpose, selected LAB were suspended in Ringer solution (Scharlab S.L., Sentmentat, Spain) at a 0.5 dilution factor according to the McFarland scale (Biomérieux S.A., France), which corresponds to 1.5 × 10^8^ colony-forming units (CFU)/mL. Then, 100 μL of the bacterial suspensions was streaked on Mueller Hinton agar plates. Antibiotic disks were then placed on the agar by gently pressuring them. Plates were incubated for 24 h under the specific temperature and respiration conditions for each microorganism. Inhibition haloes around the antibiotic disks were measured to establish the resistance/susceptibility to the above-mentioned antibiotics. 

#### 2.3.2. Evaluation of Microbial Resistance to Gastrointestinal Conditions

Strains that were found to be sensitive to the action of the antimicrobial agents were selected to evaluate their resistance to the passage through the gastrointestinal tract. In brief, selected LAB strains were subjected to different concentrations of lysozyme (Sigma-Aldrich Co., St. Louis, MO, USA), bile salts (Sigma-Aldrich Co., USA) and hydrogen peroxide (Sigma-Aldrich Co., USA), as well as to acidic pH (Panreac Química S.A., Barcelona, Spain) [23] in a micro-well plate. Resistance to lysozyme was evaluated by facing the selected strains to three different increasing lysozyme concentrations (100 μg/mL, 200 μg/mL and 300 μg/mL) at 37 °C for 60 min. Regarding bile acids resistance, selected LAB strains were cultured at three different concentrations of bile salts, namely, 0.3%, 0.5% and 1% (p/v) at 37 °C for 180 min. Resistance to hydrogen peroxide was evaluated by exposing the strains to increasing concentrations of hydrogen peroxide (10 μg/mL, 20 μg/mL and 30 μg/mL) at 37 °C for 30 min. Finally, resistance to acidic pH was evaluated by submitting the selected strains to increasing pH values from 1.5 to 6.5 (0.5 intervals) at 42 °C for 120 min (Table 1).

First of all, selected LAB strains were incubated in MRS broth for 24 h ± 2 h under the specific conditions of the strain (which included both temperature and respiration). Then, 200 μL of each reagent and 20 μL of each strain were transferred into microplates. A positive control with MRS broth without reagents and a negative control with 20 μL of sterile MRS broth without any LAB strain were also included. Finally, a microplate with 220 μL of MRS broth was also included for background measurement. Microplates were then incubated under the conditions specified in Table 1. After incubation, samples were evaluated by means of a spectrophotometer Heales MB-580 (Heales, Shenzhen, China) at 620 nm wavelength to determine the concentration of LAB and calculate the CFU.

Assays were carried out in triplicate.

#### 2.3.3. Evaluation of Antimicrobial Capacity

The LAB strains that showed microbial safety and resistance to gastrointestinal conditions were further evaluated for their antimicrobial capacity. For that purpose, and based on the Kirby Bauer method, the selected LAB were faced with 12 microorganisms, either potentially pathogenic or common environmental contaminants: *Kocuria rhizhopila*, *Pseudomonas aeruginosa*, *Bacillus subtilis*, *Bacillus cereus*, *Escherichia coli*, *Proteus* spp., *Staphylococcus aureus*, *Enterococcus faecalis*, *Streptococcus* spp., *Salmonella tiphymurium*, *Listeria monocytogenes* and *Listeria innocua* [24,25]. Two different modifications of the Kirby Bauer method were performed.

In the first modification, a 0.5 McFarland (Biomérieux S.A., France) dilution of each pathogenic microorganism was carried out and plated on Muller Hinton agar (problem plates). Then, selected LAB strains were diluted at 0.5 McFarland, and sterile antimicrobial susceptibility disks (Oxoid, Basingstoke, UK) were moistened in the dilution and deposited on the plates streaked with the above-described pathogenic/contaminant microorganisms. A positive control for both potentially pathogenic and LAB microorganisms was performed by inoculating 100 μL of the 0.5 McFarland dilutions in Mueller Hinton agar plates.

Plates were then incubated at 37 °C ± 1 °C for 24 h ± 2 h at their respective breathing and temperature conditions. Finally, the possible inhibition haloes produced by the antimicrobial capacity of LAB were measured.

In the second modification, sterile antimicrobial susceptibility disks were replaced by agar disks with LAB growth. For that purpose, selected strains of LAB were diluted with McFarland (dilution factor: 0.5) and 100 μL was streaked on MRS agar plates. After a 24 h ± 2 h period of incubation under the corresponding temperature and respiration conditions, an agar disk was obtained by means of a sterile punch and deposited on the problem plates, which had been previously inoculated with the different pathogenic microorganisms. Plates were then incubated for 24 h ± 2 h at 37 °C ± 1 °C, and inhibition haloes were further read.

### 2.4. Quantification of Vaginal LAB

Quantification of vaginal LAB was performed directly on the streaked plate by means of a colonies counter. Four categories were set: 0 (0–30 CFU/mL), 1 (>30–300 CFU/mL), 2 (>300–1000 CFU/mL) and 3 (>1000 CFU/mL).

### 2.5. Statistical Analysis

Data analysis was performed using a statistical package (IBM^®^ SPSS^®^ for Windows 27.0; Armonk, NY, USA). Percentages of vaginal LAB from bitches included in the study were compared through a chi-square test and Z-test with Bonferroni correction. The level of significance was set at *p* ≤ 0.05.

## 3. Results

### 3.1. LAB Strains Identifications

Of the 94 bitches, 37 yielded negative cultures for LAB, including those with a counting <10 CFU, thus rendering a total of 60.6% of bitches yielding positive growth for LAB. A total of 100 LAB strains were initially isolated from the 57 positive swabs. The most frequently isolated lactic acid-producing bacteria was *Lactococcus* spp. (51% of samples), followed by *Lactobacillus* spp. (25% of samples) (Table 2). After applying catalase and oxidase tests, and evaluating haemolytic capacity, 13 LAB strains were selected and identified by means of the API^®^ 50 CHL miniaturized system (Table 3). Strains belonging to the same species were differently numbered for their identification.

### 3.2. Evaluation of Probiotic Potential

#### 3.2.1. Evaluation of Microbial Safety

Regarding sensitivity to antibiotics, only six LAB, specifically Lactococcus lactis 1, Lactobacillus curvatus 1, Lactobacillus plantarum 1, Lactobacillus plantarum 2, Lactobacillus delbrueckii and Lactobacillus plantarum 3, fulfilled the requirements specified in the “Guidance on the characterization of microorganisms used as feed additives or as production organisms” [25] (Table 4).

#### 3.2.2. Evaluation of Microbial Resistance to Gastrointestinal Conditions

Once results for antimicrobial sensitivity were obtained, only the six LAB strains meeting the EFSA [23,26] requirements were subjected to the analysis of resistance to gastrointestinal conditions. Strains had to be able to grow at concentrations >300 μg/mL of lysozyme, 0.3% of bile salts, >30 μg/mL of hydrogen peroxide and pH resistance between 2 and 4. All the six selected LAB strains showed to be resistant to the established gastrointestinal conditions (Table 5).

#### 3.2.3. Evaluation of the Antimicrobial Activity of LAB Strains

Results of the evaluation of the antimicrobial activity of the selected LAB strains are summarized in Table 6. Focusing on the applied diffusion technique, the first modification of the Kirby Bauer method showed more cases of microbiological resistance to LAB than the second modification. Focusing on this second modification, only *L. lactis*, *L. plantarum* 1 and *L. plantarum* 2 showed antimicrobial activity against all the tested microorganisms. *L. lactis* 1 did not inhibit the growth of *Bacillus subtilis, Escherichia coli, Proteus* spp. and *Bacillus cereus* in the first modification, whereas both *L. plantarum* 1 and *L. plantarum* 2 were not able to inhibit the growth of *Salmonella tiphymurium*.

### 3.3. Quantification of Vaginal LAB

As far as quantification of LAB is concerned, neutered bitches showed a significantly higher percentage of bitches without bacterial growth (i.e., category 0) compared to the other groups. No significant differences between groups, however, were observed in category 3 (Table 7).

## 4. Discussion

The interest in the use of microorganisms as probiotics has progressively increased over the years, those producing lactic acid being the most commonly used [26]. Because it is essential to introduce microorganisms that do not alter the resident microbiota, the specific requirements of the host species have to be considered to improve the benefits of the probiotic agent [23,27]. Based on this assumption, several species-specific probiotics have been developed in humans [28], cattle [29] and poultry [30], among others. Studies focused on probiotics of canine origin, nevertheless, are not so abundant and are mainly centred on probiotics from a faecal source [31,32].

Regarding studies focused on probiotics from a vaginal origin and their potential benefits, while they have been widely reported in the literature in the case of women, they are very scarce in bitches [4].

In the present study, 60.6% of bitches yielded a positive culture for LAB, being very similar to previous studies which obtained 59% of positive cultures [4]. The present results, however, disagree with others obtained previously [33,34]. Thus, whereas Hutchins et al. [33] obtained positive cultures for LAB in 20% of the evaluated bitches, Golińska et al. [34] observed a total absence of LAB in vaginal swabs from bitches, either healthy or with vaginitis. The difference in the obtained percentages between the present study and that of Hutchins et al. [33] could be explained by the fact that only spayed bitches were included in the latest. According to the present results, spayed bitches yielded a lower percentage of positive cultures for LAB. On the other hand, Golińska et al. [34] transferred the sampling swabs to 1 mL of Schaedler’s broth and, after agitation, decimal dilutions were performed. So, another feasible explanation could be the low concentration of LAB in the samples that could not have yielded a significant growth once streaked on the MRS agar plates.

The most prevalent species isolated in this study were those pertaining to *Lactococcus* genera, which disagrees with previous studies where the main species belonged to *Lactobacillus* genera [4]. In that specific study, bitches with different systemic pathologies were included, whereas in the present study, only healthy or bitches with vaginitis were included. Vaginal LAB have been demonstrated to be of faecal origin [35]. Thus, one could reasonably assume that a systemic pathology able to modify the composition of intestinal microbiome could, in turn, modify the composition of vaginal microbiome. This hypothesis is sustained by a previous study [36] which demonstrated that oral supplementation with *Lactobacillus reuteri* and *Lactobacillus fermentum* significantly increased the prevalence of this specific LAB in the vaginal lumen of bitches.

*Lactobacillus* spp. is known to inhibit the growth of potential pathogens by means of producing lactic acid, bacteriocins and hydrogen peroxide [37,38,39]; these bacteria are generally considered to be safe and are widely used in fermented foods. Specifically focusing on *L. plantarum*, this LAB is commonly used for foodstuff fermentation processes [40] and several studies have demonstrated its role in gut health, metabolic disorders and mental health (see [41] for a review).

In order for LAB to be considered as probiotic agents, they have to fulfil specific requirements such as being negative to catalase and oxidase tests, positive β-haemolytic and antimicrobial activity, microbiologically safe and resistant to gastrointestinal conditions. In the present study, 13 out of the 100 obtained LAB strains were initially selected as potential in vitro probiotics after performing catalase, oxidase and haemolysis tests.

Despite being considered as beneficial, LAB are still microorganisms, which means they cannot withstand antimicrobial drugs if they are aimed to be used as probiotic agents. Resistance to antibiotics has been observed in some commercial probiotic strains [42] and specific resistance to vancomycin has been demonstrated to be chromosomally encoded in some *Lactobacillus* species [43]. Thus, probiotic agents must be sensitive to antimicrobial drugs to avoid the transference of resistance gens when administered. In the present study, only *Lactococcus lactis* 1, *Lactobacillus curvatus* 1, *Lactobacillus plantarum* 1, *Lactobacillus plantarum* 2, *Lactobacillus delbrueckii* and *Lactobacillus plantarum* 3 showed sensitivity to the whole set of antibiotics tested. According to the sensitivity test results, these strains should be considered as safe under in vivo conditions. These results reinforce the necessity of evaluating each LAB strain before accepting it as a probiotic agent, because some of them are resistant to common antimicrobial drugs.

Being resistant to gastrointestinal conditions is also a mandatory requirement for a probiotic candidate. The acid pH of the stomach, together with the antimicrobial action of pepsin, are efficient barriers to microorganisms’ survival. In addition, the tolerance to bile salts is required for survival in the small intestine [44]. In the present study, the six strains subjected to gastrointestinal conditions—in terms of acid pH, bile salts, H_2_O_2_ and lysozymes—were able to survive and grow, suggesting that they are likely to survive the passage through the gastrointestinal tract in in vivo conditions.

Finally, another condition for a microorganism to be considered as a probiotic is antimicrobial activity. In the present study, two different modifications of the Kirby Bauer method were performed to evaluate such activity. The main difference between both modifications resides in the way LAB were exposed to the pathogen microorganism. In the first modification, commercial susceptibility diffusion disks were soaked in a 0.5 McFarland medium dilution of LAB while, in the second modification, susceptibility disks were replaced by agar disks with LAB growth. The first modification allowed evaluating the activity of the released metabolites from LAB against pathogens, whereas the second one allowed assessing the action of LAB themselves against the pathogen microorganism.

According to the present results, better antimicrobial capacities were observed with the second modification, suggesting that the selected LAB microorganisms show better antimicrobial activity than their metabolites. On the other hand, only three strains, namely *L. lactis* 1, *L. plantarum* 1 and *L. plantarum* 2, showed antimicrobial activity against all tested pathogen microorganisms. While the in vitro probiotic capacity of these strains has been demonstrated, in vivo studies are needed to establish their actual probiotic potential.

Quantification of LAB has yielded remarkable results in the present study. The percentage of bitches with low quantification or even absence of vaginal LAB was significantly higher in neutered bitches than in the other groups. It is remarkable that, although not statistically significant, neutered bitches also showed a higher percentage of animals in category 0 than heat and pregnant females. Probably, a higher number of vaginitis individuals would render significant differences in neutered bitches compared to heat and pregnant bitches. It is important to highlight that the pregnant group was composed only of five individuals, which is a limitation of the present study.

The role of LAB in vaginal health has been widely studied in women. These acid-producing bacteria constitute the first line of defence against opportunistic pathogens in the vaginal lumen. Thus, although studies focused on bitches are scarce [4,33,34], a probable role of these bacteria in canine vaginal health is assumable. This hypothesis is supported by our results, which showed that 50% of bitches presenting clinical vaginitis yielded very low counts or even absence of LAB in the vaginal mucosa. Nevertheless, the presence of vaginitis, even at high concentrations of LAB, leads us to conclude that other defence mechanisms must be involved in vaginal health. In women, surfactant protein A has been shown to provide host defence in the vagina by means of different mechanisms such as pathogen opsonization, alteration of pro-inflammatory cytokine levels, stimulation of oxidative burst and promotion of antigen-presenting cells [45], the combination of both SP-A and vaginal microbiome being responsible for keeping vaginal infections under control [46]. Thus, other defence mechanisms in front of vaginal infections, in addition to LAB, are presumable also in the bitch. This would explain the presence of vaginitis in bitches that yielded positive cultures for LAB.

Pyometra is a common reproductive problem in bitches. Pathogenesis of pyometra is not yet well understood, but both hormonal and bacterial factors have been suggested to play a role. The most commonly isolated bacteria in pyometra is *Escherichia coli*, although some other microorganisms have been demonstrated to be involved (see [47] for a review). In this scenario, and because *E. coli* is a natural inhabitant of the vaginal microbiome, pyometra is generally accepted as a potential ascending infection. It thus seems reasonable to surmise that LAB could also play a role in pyometra; however, this hypothesis needs further research to be substantiated.

Another interesting result concerns neutered bitches. In the present study, nearly 70% of the evaluated bitches showed very low counts or absence of vaginal LAB. This result agrees with those previously observed in women. As stated above, lactobacilli are the most prevalent bacteria in women from puberty until the onset of menopause, strongly suggesting a link with oestrogen levels (see [46] for a review). Neutered bitches are comparable to menopausal women, since no ovarian steroids are produced. Thus, low or inexistent levels of vaginal LAB could be somehow expectable in neutered bitches. Regarding neutered bitches with high levels of LAB, a possible explanation could be the lapse of time between neutering and sampling. All these bitches had been neutered at least two months before sampling, but an actual recording of how long they had been neutered was not available. Furthermore, the required period for the vaginal microbiome to switch once the effect of oestrogen has been removed is not known. Thus, we can only hypothesize that bitches with shorter periods between neutering and sampling would not have had enough time to modify the composition of their vaginal microbiome. This, however, is just a hypothesis that needs further research.

## 5. Conclusions

Only 3 out of 100 isolated strains of LAB, all of them belonging to the genus *Lactobacillus*, showed in vitro probiotic potential according to their microbial safety, resistance of gastrointestinal conditions and antimicrobial capacity, *L. plantarum* 1 and *L. plantarum* 2 being those that showed the highest antimicrobial capacity against potentially pathogenic microorganisms. In vivo studies, therefore, are warranted to establish their potential probiotic activity before they can be considered for clinical use.

On the other hand, the present results highlight the relevance of not considering every LAB strain as a potential probiotic agent. As demonstrated herein, only few strains fulfilled the specific requirements, which supports that an individual evaluation of their probiotic capacity is mandatory.

## Figures and Tables

**Table 1 animals-12-00796-t001:** Concentration of reagents and incubation conditions for the evaluation of the resistance of lactic strains to gastrointestinal conditions [23].

Reagents for the Test of Resistance to Gastrointestinal Conditions				
	Lysozyme	Bile Salts	Hydrogen Peroxide	Acidic pH (Adjusted with HCl 1 M)
Concentration ranges	100 μg/mL	0.3% p/v	10 μg/mL	1.5 to 6.5 (at 0.5 intervals)
	200 μg/mL	0.5% p/v	20 μg/mL	
	300 μg/mL	1% p/v	30 μg/mL	
Incubation conditions	37 °C	37 °C	37 °C	42 °C
	60 min	180 min	30 min	120 min

**Table 2 animals-12-00796-t002:** Lactic acid-producing bacteria isolated from canine vaginal swabs.

Microorganism	n (%)
*Lactococcus* spp.	51
*Lactobacillus* spp.	24
*Pediococcus acidilactici*	9
*Lactobacillus plantarum*	4
*Lactococcus lactis*	3
*Lactobacillus brevis*	2
*Lactobacillus curvatus*	2
*Lactobacillus delbrueckii* spp. *delbrueckii*	1
*Lactobactillus paracasei*	1
*Lactobactillus rhamnosus*	1
*Bifidobacterium* spp.	1
*Leuconostoc* spp.	1

**Table 3 animals-12-00796-t003:** Selected LAB strains with α and γ haemolytic capacity, and negative catalase/oxidase tests.

Isolates Identify	Haemolysis	Catalase	Oxidase
*Lactobacillus brevis* =	γ haemolytic	-	-
*Lactobacillus curvatus* 1	α haemolytic	-	-
*Lactobacillus curvatus* 2	α haemolytic	-	-
*Lactobacillus curvatus* 3	α haemolytic	-	-
*Lactobacillus delbrueckii* spp. *delbrueckii*	γ haemolytic	-	-
*Lactococcus lactis* 1	α haemolytic	-	-
*Lactococcus lactis* 2	α haemolytic	-	-
*Lactobacillus paracasei* spp. *paracasei*	γ haemolytic	-	-
*Lactobacillus plantarum* 1	α haemolytic	-	-
*Lactobacillus plantarum* 2	α haemolytic	-	-
*Lactobacillus plantarum* 3	α haemolytic	-	-
*Lactobacillus rhamnosus*	γ haemolytic	-	-
*Pediococcus acidilactici*	α haemolytic	-	-

**Table 4 animals-12-00796-t004:** Evaluation of sensitivity to antibiotics.

Isolates Identity	AM	K	CD	VA	S	TE	GN	E	C
*Lactobacillus brevis*	S	R	R	nr	R	S	R	S	S
*Lactobacillus curvatus* 1	S	S	S	nr	S	S	S	S	S
*Lactobacillus curvatus* 2	S	R	R	nr	R	S	R	R	R
*Lactobacillus curvatus* 3	S	R	R	nr	R	S	R	R	R
*Lactobacillus delbrueckii spp delbrueckii*	S	S	S	S	S	S	S	S	S
*Lactococcus lactis* 1	S	S	S	S	S	S	S	S	S
*Lactococcus lactis* 2	S	R	S	R	R	S	R	S	S
*Lactobacillus paracasei spp paracasei*	S	R	S	nr	R	S	R	S	S
*Lactobacillus plantarum* 1	S	S	S	S	S	S	S	S	S
*Lactobacillus plantarum* 2	S	S	S	S	S	S	S	S	S
*Lactobacillus plantarum* 3	S	S	S	S	S	S	S	S	S
*Lactobacillus rhamnosus*	S	R	R	nr	S	S	S	S	S
*Pediococcus acidilactici*	S	R	S	nr	S	S	R	S	S

AM: ampicillin, K: kanamycin; CD: clindamycin; VA: vancomycin; S: streptomycin; TE: tetracycline; GN: gentamicin; E: erythromycin; C: chloramphenicol; nr: not required; S: sensitive; R: resistant.

**Table 5 animals-12-00796-t005:** Evaluation of microbial resistance to gastrointestinal conditions.

	Lysozyme	Bile Salts	Hydrogen Peroxide	pH
	(μg/mL)	(% p/v)	(μg/mL)	(HCl 1 M)
*L. delbrueckii*	>300	0.5–1	>30	<2
*L. curvatus*	>300	>1	>30	<2
*L. lactis* 1	>300	>1	>30	<2
*L. plantarum* 1	>300	0.5–1	>30	<2
*L. plantarum* 2	>300	>1	>30	<2
*L. plantarum* 3	>300	>1	>30	<2

**Table 6 animals-12-00796-t006:** Results of the antimicrobial activity for LAB strains.

	*L. plantarum* 1		*L. lactis* 1		*L. plantarum* 2		*L. curvatus*		*L. delbrueckii*		*L. plantarum* 3	
	I	II	I	II	I	II	I	II	I	II	I	II
*K. rhizophila*	S	S	S	S	S	S	R	S	S	S	S	S
*P. aeruginosa*	S	S	S	S	S	S	S	S	S	S	R	S
*B. subtilis*	S	S	R	S	S	S	S	S	R	R	R	R
*E. coli*	S	S	R	S	S	S	S	S	S	S	S	S
*Proteus* spp.	S	S	R	S	S	S	R	S	S	S	S	S
*S. aureus*	S	S	S	S	S	S	R	S	S	S	S	S
*E. faecalis*	S	S	S	S	S	S	S	S	R	S	R	S
*Streptococcus* spp.	S	S	S	S	S	S	S	S	S	S	S	S
*S. tiphymurium*	R	S	S	S	R	S	S	R	S	S	S	S
*B. cereus*	S	S	R	S	S	S	S	S	R	R	R	S
*L. monocytogenes*	S	S	S	S	S	S	S	S	R	S	S	S
*L. innocua*	S	S	S	S	S	S	R	S	R	S	S	S

I: First modification of Kirby Bauer method. II: Second modification of Kirby Bauer method. S: sensitive; R: resistant. *K. rhizophila*: *Kocuria rhyzophila*; *P. aeruginosa*: *Pseudomonas aeruginosa*; *B. subtilis*: *Bacillus subtillis*; *E. coli*: *Escherichia coli*; *S. aureus*: *Staphylococcus aureus*; *E. faecalis*: *Enterococcus faecalis*; *S. tiphymurium*: *Salmonella tiphymurium*; *B. cereus*: *Bacillus cereus*; *L. monocytogenes*: *Lysteria monocytogenes*; *L. innocua*: *Lysteria innocua*; *L. plantarum*: *Lactobacillus plantarum*; *L. lactis*: *Lactococcus lactis*; *L. curvatus*: *Lactobacillus curvatus*; *L. delbrueckii*: *Lactobacillus delbrueckii*.

**Table 7 animals-12-00796-t007:** Quantification in terms of percentages of vaginal LAB from bitches included in the study (n). Females were distributed into four groups, namely neutered, in heat, pregnant and with vaginitis. Different letters mean significant differences (*p* ≤ 0.05) between columns within a given row. Different numbers mean significant differences (*p* ≤ 0.05) between rows within a given column. 0 = 0–30 CFU/mL; 1 = >30–300 CFU/mL; 2 = >300–1000 CFU/mL; 3 = >1000 CFU/mL.

	0 (n)	1 (n)	2 (n)	3 (n)
Neutered (n = 25)	68.0 (17) ^a,1^	8.0 (2) ^b,1^	4.0 (1) ^b,1,2^	20.0 (n) ^a,b,1^
Heat (n = 48)	29.2 (14) ^a,b,2^	27.1 (13) ^a,b,1,2^	2.1 (1) ^b,1,2^	41.7 (20) ^a,1^
Pregnant (n = 5)	**−** (0) ^a,3^	60.0 (3) ^b,2^	− (0) ^a,1^	40.0 (2) ^b,1^
Vaginitis (n = 16)	50.0 (8) ^a,1,2^	12.5 (2) ^b,1^	12.5 (2) ^b,2^	25.0 (4) ^a,b,1^

## Data Availability

Data are available upon request to the authors.

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
