# Peer review of "Evaluation of the Probiotic In Vitro Potential of Lactic Acid-Producing Bacteria from Canine Vagina: Possible Role in Vaginal Health"

_animals, 2022, doi:10.3390/ani12060796_

Round 1

Reviewer 1 Report

Comments attached

Author Response

Dear reviewer,

thanks for your comments. Here are our answers in the hope they fullfil your queries.

Specific comments:

L51  The vaginal microbiome has been widely studied in the bitch  - please provide examples of literature

Answer: After reviewing the literature, the authors have realized that most of the published manuscripts are focused on the hormonal and health status of the bitch and not just the commensal vaginal microbiome. The sentence has been re-written in order to solve this problem.

LL283-284  In the present study, 60.6% of bitches yielded a positive culture for LAB, being very 283 similar to previous studies which obtained 59% of positive cultures [4,33].  - This sentence is not true. In the case of the article by Huching et al. [33] probiotics were found only in 7 of 35 bitches. Please correct the errors.

Answer: Thank you for detecting the mistake. It has been corrected in the text.

LL288-290  If contamination 288 was a fact, however, all the included bitches should have yielded a positive result . This wording cannot be accepted. Contamination is not a binary thing. Contamination does not always have to be done. Please justify your statement or expand it. 

Answer: The reviewer is completely right. This sentence is not appropriate and has been removed from the text.

L297 - Please give a specific reference. The term previous study is imprecise.

Answer: Thanks for the comment. In fact, the specific study has been named in the previous sentence. The sentence has been slightly re-written with the hope of making it more understandable.

L 201 -  Please give a specific reference. The term previous study is imprecise.

Answer: A reference has been added.

LL301-304 - I think that it should be pointed out that Huching et al. [33] research shows something very different, so it is not that clear yet.

Answer: This difference has been pointed out in the text and a feasible explanation has been provided.

LL373-380- I think that the theories and conclusions described in this paragraph are far too bold and are not backed up by any literature. The bacterial flora of the uterus in bitches is poorly understood, and its regulatory mechanisms are even more so.

Answer: Thanks for your comment. Unfortunately, the lines you state here does not coincide with those in the manuscript, so I’m just assuming (focusing on your words) that you are referring to the chapter about pyometra. The authors are pretty aware that what is written in that chapter is just mere hypothesis and so we stated in it by confirming the need of further research. Regarding to conclusions, they only describe that just few LAB strains show “in vitro” probiotic capacity, making probiotic testing mandatory to determine if a specific LAB strain can be considered as a probiotic or not.

Reviewer 2 Report

we are waiting for the in vivo studies! this work may be relevant for pyometra, could you explain more the future uses of these results in the treatment of the disease?

Author Response

Dear reviewer,

thanks for your kind comments on our manuscript and also for the suggestion of evaluating the role of LAB on pyometra. It is a more than interesting suggestion

Reviewer 3 Report

This is interesting study to isolate LAB from female dogs with respect to their estrus cycle. I have a question about the dogs' condition. Did they have no medication history in past 2 month? If so, could you mention it in the "2.1. Samples collection" section? 

Author Response

Dera reviewer,

thanks for your kind comments. The information about no medication history has been added to the manuscript as it has been suggested.